# STABLE ESTIMATION OF SURVIVAL CAUSAL EFFECTS

## ABSTRACT

We study the problem of estimating survival causal effects, where the aim is to characterize the impact of an intervention on survival times, i.e., how long it takes for an event to occur. Applications include determining if a drug reduces the time to ICU discharge or if an advertising campaign increases customer dwell time. Historically, the most popular estimates have been based on parametric or semiparametric (e.g. proportional hazards) models; however, these methods suffer from problematic levels of bias. Recently debiased machine learning approaches are becoming increasingly popular, especially in applications to large datasets. However, despite their appealing theoretical properties, these estimators tend to be unstable because the debiasing step involves the use of the *inverses* of small estimated probabilities—small errors in the estimated probabilities can result in huge changes in their inverses and therefore the resulting estimator. This problem is exacerbated in survival settings where probabilities are a product of treatment assignment and censoring probabilities. We propose a covariate balancing approach to estimating these inverses directly, sidestepping this problem. The result is an estimator that is stable in practice and enjoys many of the same theoretical properties. In particular, under overlap and asymptotic equicontinuity conditions, our estimator is asymptotically normal with negligible bias and optimal variance. Our experiments on synthetic and semi-synthetic data demonstrate that our method has competitive bias and smaller variance than debiased machine learning approaches.

## 1 INTRODUCTION

The estimation of the impact of interventions on survival times is a key objective in numerous studies. This analytical approach is important in various domains, including drug efficacy's evaluation at ICU stay duration and assessment of advertising campaigns' effects on customer dwell time.

The predominant approach for assessing the impact of interventions on survival times is to employ the Cox proportional hazards model (CoxPH) (Cox, 1972; Andersen & Gill, 1982). This model estimates conditional hazard ratios, which serve as a measure of survival causal effects. However, the causal interpretation of conditional hazard ratios within the CoxPH model can be complex (Martinussen, 2022; Vansteelandt et al., 2022). Hence, there has been a growing interest in directly estimating *counterfactual survival curves* and consequently deriving survival causal effects from these estimates (Westling et al., 2023) such as the average survival effect, the residual average survival effect, and the survival quantile effect (Mansourvar et al., 2016; Mao et al., 2018). A counterfactual survival curve represents the probability of an event occurring at a specific point in time, if contrary to fact, the entire population had undergone a specific intervention, given that the event of interest has not yet occurred.

Traditional approaches for estimating counterfactual curves in survival analysis have relied on conditional parametric or semiparametric regression models (Kleinbaum & Klein, 2012; Cox, 1972), or weighted methods based on inverse probability of censoring and treatment weighting (IPW) (Robins & Rotnitzky, 1992). However, both approaches can suffer from model misspecification i.e. the model does not contain the truth. Weighted methods are also susceptible to extreme inversions (Kang et al., 2007; Kallus & Santacatterina, 2022; Kallus, 2021). In recent years, debiased machine learning approaches have gained popularity due to their appealing double robustness property which deals with model misspecification in addition to being semiparametric efficient (Kennedy, 2022). Specifically, double robustness implies that even if only certain models that need to be estimated are correctly specified, these approaches remain consistent, ie., asymptotycally unbiased.

Semiparametric efficiency refers to the ability of an estimator or estimation method to achieve the smallest possible *asymptotic* variance among a class of semiparametric estimators. Despite their desirable theoretical properties, debiased machine learning methods *still* face challenges associated with extreme inversions because they are necessarily characterized by inverse probability weights (Chernozhukov et al., 2022a).

Covariate balancing methods have been proposed as a solution to the issue of extreme weights encountered in other approaches (Ben-Michael et al., 2021). Instead of directly incorporating inverse probabilities, these methods employ a weight-learning process that yields more stable estimates while preserving crucial properties like semiparametric efficiency. However, the application of these methods to estimate survival causal effects has been significantly limited (Xue et al., 2023; Wong & Chan, 2017; Abraich et al., 2022; Johansson et al., 2022; Santacatterina, 2023; Kallus & Santacatterina, 2021; Yiu & Su, 2022), with none of them providing results on asymptotic normality. Asymptotic normality is crucial for statistical inference, hypothesis testing, and the construction of confidence intervals, highlighting a current gap in the literature.

We contribute to the literature of survival causal effects estimation by presenting a novel approach based on covariate balancing techniques. Our proposed method offers stability in practical applications, and when certain conditions such as overlap and asymptotic equicontinuity are met, it demonstrates asymptotic normality with minimal bias and optimal variance, ie., statistically efficiency. We substantiate these claims both theoretically and through experiments conducted on synthetic data. Proofs, additional experiments and some technical details are deferred to the supplemental material.

## 2 SETTING

In this section, we review the notations and main quantities in standard discrete survival analysis (Section 2.1). We refer interested readers to e.g. Chapter 16 of Crowder (2012) for further details. We then introduce counterfactual survival analysis including the causal parameter of interest and identification assumptions using the Neyman-Rubin potential outcome framework (Neyman, 1923; Rubin, 1974) (Section 2.2). Finally, some useful notations are introduced in Section 2.3.

### 2.1 DISCRETE SURVIVAL ANALYSIS

**Data structure.** We define an observable survival data unit as the set of random variables $(X, E, \widetilde{T})$ where $X \in \mathcal{X} \subseteq \mathbb{R}^d$ is the covariate recorded prior to the beginning of the study. To define $E$ and $\widetilde{T}$, let $T \in \mathcal{T} \cup \{\infty\}$ be the time-to-event and $C \in \mathcal{T}$ the time-to-censoring (censor time), where $T = \infty$ means that the event has not happened during the study time window $\mathcal{T}$. In a clinical study, $T$ can be the patient's time from their entry to the study until their death - the event of interest. The events may not happen during the study's time or the patients drop out before the end of the study, therefore a censoring time $C$ is introduced. Define $\widetilde{T} = \min\{T, C\}$ the right-censored time and $E = \mathbf{1}(T \leq C)$ the event indicator. If $E = 1$, the event is observed and occurs at time $\widetilde{T} = T$, otherwise, the event has not happened or is censored at time $\widetilde{T} = C$. We assume discrete $\mathcal{T} = [t_{max}] = \{0, 1, .., t_{max}\}$, $t_{max} \in \mathbb{Z}_+$ and $P(T = 0) = 0$, so that zero times are ruled out. Denote $|\mathcal{T}|$ the total number of time points and $|t| = |\{u \in \mathcal{T} : u \leq t\}|$ the number of time points less than or equal to $t$. $t_{max}$ may be chosen by the user, or as a result of administrative censoring $P(C \leq t_{max}) = 1$.

**The marginal and sub-survival functions.** We seek to recover the distributions of latent time $T$ from the observable $\widetilde{T}$ and $E$. Define the marginal-hazard, the marginal-survival function of $T$ and the marginal-censoring function of $C$ as follows:

$$h_t(x) = P(T = t | X = x, T \geq t); \quad S_t(x) = P(T > t | X = x); \quad G_t(x) = P(C > t | X = x). \tag{1}$$

These definitions imply a one-to-one relationship between $h$ and $S$: letting $t- = \max(0, t - 1)$,

$$h_t(x) = \frac{P(T = t | X = x)}{P(T \geq t | X = x)} = \frac{P(T = t | X = x)}{S_{t-}(x)}; \quad S_t(x) = S_{t-}(x)(1 - h_t(x)) = \prod_{u \leq t}(1 - h_u(x)). \tag{2}$$

For observable data, define the sub-hazard and sub-survival function as:

$$\lambda_t(x) = P(\widetilde{T} = t, E = 1 | X = x, \widetilde{T} \geq t); \quad H_t(x) = P(\widetilde{T} > t | X = x). \tag{3}$$

When $T$ and $C$ are conditionally independent given $X$, the following lemma relates the sub-distributions and the marginal-distributions:

**Proposition 2.1.** If $T \perp C | X$ then: $h_t(x) = \lambda_t(x)$, therefore:

$$S_t(x) = \prod_{u \leq t}(1 - h_u(x)) = \prod_{u \leq t}(1 - \lambda_u(x)). \tag{4}$$

Additionally, the sub-survival function decomposes into a product of the marginal-survival functions of the event and censoring: $H_t(x) = S_t(x)G_t(x)$.

We provide a proof in Appendix B. This equivalence enables the estimation of all identities defined up to now from observable data. From here on, we will use term hazard to refer to the sub-hazard.

## 2.2 COUNTERFACTUAL SURVIVAL ANALYSIS

**Data structure.** We define the ideal data unit in counterfactual survival analysis as $(X, E, A, T(0), T(1), C(0), C(1))$, where $A \in \{0, 1\}$ is a binary random variable indicating e.g. whether or not a patient receives the treatment, $T(a)$ is the event time of interest under $A = a$ and similarly for $C(a)$. With the introduction of $A$, we define $T = AT(1) + (1 - A)T(0)$ and $C = AC(1) + (1 - A)C(0)$, therefore the observable time $\widetilde{T}$ and event indicator $E$ are defined as before. The observable data unit is now $O = (X, E, A, \widetilde{T})$. In this context, we will talk about the treatment-specific hazard and survival function, $\lambda_t(a, x)$ and $S_t(a, x)$, characterized as in Section 2.1 in terms of the conditional distribution given treatment $A = a$. With Lemma 2.1, we then define,

$$\lambda_t(x, a) = P(E = 1, \widetilde{T} = t | X = x, A = a, \widetilde{T} \geq t); \quad S_t(x, a) = \prod_{u \leq t}(1 - \lambda_u(x, a)) \tag{5}$$

**Survival causal parameter of interest: the counterfactual survival function.** We will focus mainly on the counterfactual survival function at time $t \in \mathcal{T}$ and treatment $a \in \{0, 1\}$: $\psi^{a,t} = P(T(a) > t)$. Other commonly encountered parameters can be built from it such as the average survival effect at time $t$: $(\psi^{1,t} - \psi^{0,t})$ and the treatment-specific mean survival time $\sum_{t \leq t_{max}} \psi^{a,t}$ as well as its average effect counterpart. Where convenient, we will write $\psi$ in place of $\psi^{a,t}$, letting the treatment and time of interest be inferred from context.

**Identification** To identify the counterfactual survival function using the observable data, similar to Hubbard et al. (2000); Bai et al. (2013; 2017); Westling et al. (2023); Díaz (2019); Cai & van der Laan (2020), we require the following testable and untestable assumptions:

- (A1) $T(a), C(a) \perp A | X$ for each $a \in \{0, 1\}$.
- (A2) $T(a) \perp C(a) | A = a, X$ for each $a \in \{0, 1\}$.
- (A3) $P(A = a | X) = 0 > 0$ almost surely.
- (A4) $P(C(a) \geq \tau | X) > 0$ positivity (censoring),

**Proposition 2.2.** When Assumptions (A1)-(A4) hold, $\psi^{a,t}$ can be computed by observable quantities

$$\psi^{a,t} = \mathbb{E}\left[S_t(X, a)\right] = \mathbb{E}\left[\prod_{u \leq t}(1 - \lambda_u(X, a))\right]. \tag{6}$$

The proof, along with a discussion of the assumptions, appears in Appendix B.

### 2.3 Notation for our proposed approach

In the next section, it will be useful to think of the hazard at time $u$, $\lambda_u(a, x)$, as $\bar{\lambda}_u(x, a, 1)$ where

$$
\begin{aligned}
\bar{\lambda}_u(x, a, g) &= P(E = 1, \widetilde{T} = u | X = x, A = a, 1(\widetilde{T} \geq u) = g_u) \\
&= \mathbb{E}[Y^u \mid X = x, A = a, \mathbf{1}(\widetilde{T} \geq u) = g_u] \qquad \text{for} \quad Y^u = \mathbf{1}(E = 1, \widetilde{T} = u).
\end{aligned}
\tag{7}
$$

Furthermore, we'll think of the functions $\bar{\lambda}_1 \dots \bar{\lambda}_T$ as the *components* of a vector-valued function $\bar{\lambda}$ with $[\bar{\lambda}(x, a, t)]_u = \bar{\lambda}_u(x, a, 1(t \geq u))$ Note that this vector-valued function has the property that its $u$th component depends on $t$ *only* through the indicator $1(t \geq u)$; we will call the space of vector-valued functions with this property the 'hazard-like functions' $\Lambda$. And we will think of our estimand $\psi^{a,t}$, as identified in Proposition 2.2, as a functional $\psi$ on this space $\Lambda$ *evaluated at* $\bar{\lambda}$:

$$
\psi^{a,t} = \psi(\bar{\lambda}) \quad \text{where} \quad \psi(\bar{l}) = \mathbb{E}\left[ \prod_{u \leq t} \left( 1 - \bar{l}_u(X, a, 1) \right) \right].
\tag{8}
$$

We've written things in these terms to emphasize the analogy between our estimation problem and the problem of estimating a functional of a conditional expectation function considered in Hirshberg & Wager (2021) and Chernozhukov et al. (2022b). What we are estimating is a functional of a vector-valued function $\bar{\lambda} \in \Lambda$ that is, in each component, the conditional expectation function of an observed outcome $Y^u$ given observed conditioning variable $\{X, A, 1(\widetilde{T} \geq u)\}$. We will work with an inner product on this space $\Lambda$ defined in terms of a random variable $X, A, \widetilde{T}$ with the same distribution as $(X_1, A_1, \widetilde{T}_1) \dots (X_n, A_n, \widetilde{T}_n)$.[1] Letting $G^u = 1(\widetilde{T} \geq u)$ here and below,

$$
\langle f, g \rangle = \mathbb{E}\left[ \sum_{u \leq t} f_u(X, A, G^u) g_u(X, A, G^u) \right] \quad \text{for} \quad f, g \in \Lambda.
\tag{9}
$$

While we cannot evaluate the functional $\psi$ or the inner product $\langle \cdot, \cdot \rangle$ exactly because we do not know the distribution of $(X, A, \tilde{T})$, natural sample-average approximations are available.

$$
\psi_n(\bar{l}) = \frac{1}{n} \sum_{i=1}^{n} \left[ \prod_{u \leq t} \left( 1 - \bar{l}_u(X_i, a, 1) \right) \right] ; \quad \langle f, g \rangle_n = \frac{1}{n} \sum_{i=1}^{n} \left[ \sum_{u \leq t} f_u(X, A, G^u) g_u(X, A, G^u) \right]
\tag{10}
$$

## 3 Approach

### 3.1 A first-order approximation and the derivative's Riesz representer

Given an estimate $\hat{\lambda}$ of the hazard $\bar{\lambda}$, which we can get e.g. by using a machine-learning method of our choice to regress $Y^u$ on $X, A, G^u$ at each timestep $u$, the 'plug-in estimate' $\psi_n(\hat{\lambda})$ is a natural estimate of $\psi(\bar{\lambda})$. But we can improve on this using a *first-order correction*. Consider the first-order Taylor expansion of $\psi$ around $\hat{\lambda}$. In terms of its derivative $d\psi(\hat{\lambda})(h)$ at $\hat{\lambda}$ in the direction $h$,

$$
\psi(\hat{\lambda} + h) \approx \psi(\hat{\lambda}) + d\psi(\hat{\lambda})(h) \quad \text{for } h \in \Lambda.
\tag{11}
$$

Taking $h = \bar{\lambda} - \hat{\lambda}$, the difference between our actual hazard and our estimate, we get a first-order approximation to our estimand $\psi(\bar{\lambda})$. Our main concern will therefore be the estimation of this derivative term $d\psi(\hat{\lambda})(\lambda - \hat{\lambda})$ where, as established in Lemma C.2 in the appendix,

$$
d\psi(\hat{\lambda})(h) = \mathbb{E}\left[ \sum_{u \leq t} \hat{r}_u(X, a) h_u(X, a, 1) \right] \quad \text{for} \quad \hat{r}_u(X, a) = -\hat{S}_t(X, a) \frac{\hat{S}_{u-}(X, a)}{\hat{S}_u(X, a)}
\tag{12}
$$

---

[1]Note that this is an average over $X, A, \widetilde{T}$ *only*. If $\hat{f}$ and $\hat{g}$ are random variables, $\langle \hat{f}, \hat{g} \rangle_t$ will be the random variable $\mathbb{E}[\sum_{u \leq t} \hat{f}_u(X, A, G^u) \hat{g}_u(X, A, G^u) \mid \hat{f}, \hat{g}]$. We will do this conditioning implicitly throughout.

**While $\hat{\lambda}$ and $\hat{r}$ are known quantities, derived from our hazard estimate $\hat{\lambda}$, we will need to substitute something for the *actual* hazard $\bar{\lambda}$.** For this purpose, $Y_u$ is a natural proxy, as its conditional mean *is* the hazard $\bar{\lambda}_u(X, A, G)$ at the observed levels of the conditioning variables $X, A, G$; our challenge is make this speak to its value $\bar{\lambda}_u(X, a, 1)$ at often-unobserved levels of those variables. To this end, we observe that the Riesz representation theorem guarantees that, for the space $\Lambda$, there exists a unique element $\gamma$ that acts (via an inner product) on the function $h$ like the functional derivative does, i.e., one satisfying

$$d\psi(\hat{\lambda})(h) = \langle \gamma, h \rangle = \sum_{u \leq t} \mathbb{E}\left[\gamma_u(X, A, G)h(X, A, G)\right] \quad \text{for all} \quad h \in \Lambda \tag{13}$$

and, in particular, taking $h = \bar{\lambda} - \hat{\lambda}$,

$$d\psi(\hat{\lambda})(\lambda - \hat{\lambda}) = \langle \gamma, \lambda - \hat{\lambda} \rangle = \sum_{u \leq t} \mathbb{E}\left[\gamma_u(X, A, G)\left\{\bar{\lambda}(X, A, G) - \hat{\lambda}(X, A, G)\right\}\right]. \tag{14}$$

We call $\gamma$ the *Riesz representer* of $d\psi(\hat{\lambda})$. Substituting $Y_u$ for $\bar{\lambda}$ gives us an equivalent expression for our derivative term. By the law of iterated expectations,

$$\begin{aligned}
d\psi(\hat{\lambda})(\lambda - \hat{\lambda}) &= \sum_{u \leq t} \mathbb{E}\left[\gamma_u(X, A, G_u)\left\{\mathbb{E}[Y_u | X, A, G_u] - \hat{\lambda}_u(X, A, G_u)\right\}\right] \\
&= \sum_{u \leq t} \mathbb{E}\left[\gamma_u(X, A, G_u)\left\{Y_u - \hat{\lambda}_u(X, A, G)\right\}\right]
\end{aligned} \tag{15}$$

So, supposing that we have an estimator $\hat{\gamma}$ for our Riesz representer, we obtain the following estimator for $\psi(\bar{\lambda})$ by replacing expectations with sample averages as in Equation (10).

$$\hat{\psi} = \hat{\psi}_n(\hat{\lambda}) + \frac{1}{n}\sum_{i=1}^{n}\sum_{u \leq t} \hat{\gamma}_u(X_i, A_i, G_i)\left\{Y_i^u - \hat{\lambda}_u(X_i, A_i, G_i)\right\} \tag{16}$$

### 3.2 Estimating the Riesz Representer $\gamma$

$\gamma$ can be characterized in terms of *inverse probability weights*. The explicit solution to the set of equations defining the Riesz representer (Equation 13) is

$$\begin{aligned}
\gamma(X, A, G) &= \frac{\hat{r}_u(X, a)\mathbf{1}(A = a, G_u = 1)}{\mathbb{E}\left[\mathbf{1}(A = a, G_u = 1)|X\right]} = \hat{r}_u(X, a)\mathbf{1}(A = a, G_u = 1)\omega_u(X, a) \\
&\text{where} \quad \omega_u(x, a) = \frac{1}{H_{u-}(x, a)\pi(x, a)} \text{ and } \pi(x, a) = P(A = a \mid X = x).
\end{aligned} \tag{17}$$

One approach is to estimate the functions $\pi(x, a)$ and $H_{u-}(x, a)$ appearing in Equation (17) and assemble them into an estimate $\hat{\gamma} = \frac{\hat{r}_u(X,a)\mathbf{1}(A=a,G_u=1)}{\hat{H}_{u-}(X,a)\hat{\pi}(X,a)} = \hat{r}_u(X, a)\mathbf{1}(A = a, G_u = 1)\hat{\omega}_u(X, a)$ of the Riesz representer $\gamma$. Taking this approach yields 'one-step' estimator discussed in Section 4 below. However, this solution suffers from instability, common in all inverse weight-based estimators. When the ground truth functions $H_{u-}(X, a)$ and $\pi(X, a)$ are very small at some observations, naturally their estimators tends to be very small, hence, slight errors in estimating them can result in large errors in the estimation of their inverses (i.e. $1/(x - \epsilon) - 1/x \approx \epsilon/x^2$). Thus, lack of overlap in the data means that such estimators will be unstable, with very large sample sizes needed to get estimates of $\pi$ and $H$ accurate enough to tolerate inversion. Moreover, even when when overlap in the data is not poor, moderate-sized errors in the estimation of $\pi$ and $H$ that occur at smaller sample sizes results in similar issues. It is common practice to clip these weights to a reasonable range before using them, but ad-hoc clipping often results in problematic levels of bias. In short, we lack practical and theoretically sound methods to make this inversion step work reliably.

Our approach, which avoids this problematic inversion, focuses not on the analytic form of the Riesz representer but on what it does, i.e., on the equivalence of the inner product $\langle \gamma, h \rangle$ to the derivative evaluation $d\psi(\hat{\lambda})(h)$. As we lack the ability to analytically evaluate the expectations involved in both, we will use the sample average approximations of these quantities: $\langle \gamma, h \rangle_n$ (defined above) and $d\psi_n(\hat{\lambda})(h) = \frac{1}{n}\sum_{i=1}^{n}\sum_{u \leq t} \hat{r}_u(X_i, a)h_u(X_i, a, 1)$. Inspired by the explicit characterization in

Equation (17), taking $\hat{\gamma}$ to have a functional form $\hat{\gamma}(X_i, A_i, G_i) = \hat{r}_u(X_i, a)\mathbf{1}(A_i = a, G_{iu} = 1)\hat{\omega}_{iu}$ (for weights $\hat{\omega}_{iu}$) , we ask that, for a set of functions $h$,

$$\frac{1}{n}\sum_{i=1}^{n}\sum_{u\leq t}\hat{r}_u(X_i, a)\mathbf{1}(A_i = a, G_{iu} = 1)\hat{\omega}_{iu}h_u(X_i, A_i, G_{iu}) \approx \frac{1}{n}\sum_{i=1}^{n}\sum_{u\leq t}\hat{r}_u(X_i, a)h_u(X_i, a, 1).$$
(18)

In the simpler setting considered in Hirshberg & Wager (2021), this approximation has meaning beyond that, suggested by its relationship to the population analog Equation (13). The quality of the *in-sample approximation* described by Equation (18) for the specific function $\hat{h} = \bar{\lambda} - \hat{\lambda}$ is, along with the accuracy of the estimator $\hat{\lambda}$ and therefore of the linear approximation in Equation (12), one of two essential determinants of the estimator's bias. See Appendix C.2, where we include an informative decomposition of our estimator's error and further discussion.

In light of that, we generalize the approach of Hirshberg & Wager (2021) for estimating $\gamma$ by ensuring that Equation (18) holds for a set $\mathcal{M}$ of hazard-like functions $h$, which we deem as a *model* for function $\hat{h} = \lambda - \hat{\lambda}$. In particular, we ask for weights that are (i) not too large, to control our estimator's variance, and (ii) ensure the approximation in Equation (18) is accurate uniformly over model $\mathcal{M}$. This modeling task is somewhat simplified by the observation that we *only* need a model for $\hat{h}(X_i, a, 1)$, as the presence of the indicator $\mathbf{1}(A_i = a, G_{iu} = 1)$ on the right side of Equation (18) justifies the substitution of $h_u(X_i, a, 1)$ for $h_u(X_i, A_i, G_i)$. Thus, choosing a norm $\|\cdot\|$ and taking the set of vector-valued functions $[h_1(x) \ldots h_n(x)]$ with $\sum_u \|h_u\|^2 \leq 1$ as our model for $[\hat{h}_1(X, a, 1) \ldots \hat{h}_n(X, a, 1)]$, we choose weights by solving the following optimization problem

$$\hat{\omega} = \underset{\omega \in \mathbb{R}^{n|\mathcal{T}|}}{\operatorname{argmin}} \left\{ I(\omega)^2 + \frac{\sigma^2}{nt}\sum_{i=1}^{n}\sum_{u\leq t}\hat{r}_u(X_i, a)^2\mathbf{1}(A_i = a, G_{ui} = 1)\omega_{iu}^2 \right\} \quad \text{where}$$

$$I(\omega) = \max_{\sum_{u\leq t}\|h_u\|^2 \leq 1} \frac{1}{n}\sum_{i=1}^{n}\sum_{u\leq t}\left\{\hat{r}_u(X_i, a)h_u(X_i) - \hat{r}_u(X_i, a)\mathbf{1}(A_i = a, G_{iu} = 1)\omega_{iu}h_u(X_i)\right\}.$$
(19)

What remains is to choose this norm with the intention $\|\hat{h}_u\|$ is small for all $u$. If our model is correct in the sense that $\sum_u \|\hat{h}_u\|^2 \leq B^2$, then $B$ times the maximal approximation error $I(\hat{\omega})$ bounds the approximation error in Equation (18).

As usual, there is a natural trade-off in choosing this model—if we take it to be too small, $\|\hat{h}_u\|$ will be large or even infinite; on the other hand if we take it to be too large, we will be unable to find weights for which the approximation Equation (18) is highly accurate for all functions in the model. Choosing a norm with a unit ball that is a *Donsker class*, e.g. a Reproducing Kernel Hilbert Space (RKHS) norm like the one we use in our experiments, is a reasonable trade-off that is common in the literature on minimax and augmented minimax estimation of treatment effects (e.g., Hirshberg et al., 2019; Kallus, 2016). When we do this, our estimator will be asymptotically efficient, i.e. asymptotically normal with negligible bias and optimal variance, if the hazard functions $f(x) = \bar{\lambda}_u(x, a, 1)$ are in this class and we estimate them via empirical risk minimization with appropriate regularization. We discuss the computational aspects of this problem in Appendix A.2.

### 3.3 ASYMPTOTIC EFFICIENCY

In this section, we will discuss the asymptotic behavior of our estimator, giving sufficient conditions for it to be asymptotically efficient. Throughout, we will assume $\hat{\lambda}$ is cross-fit, i.e., fit on an auxilliary sample independent of and distributed like our sample.

Our first condition is that it converges faster than fourth-root rate. This ensures the error of our first-order approximation Equation (11), quadratic in the error $h = \bar{\lambda} - \hat{\lambda}$, is asymptotically negligible.

**Assumption 3.1.** $\left\|\hat{\lambda}_u(\cdot, a) - \lambda_u(\cdot, a)\right\|_{L_2(P)} = o_p(n^{-1/4})$ for all $u \leq t$.

Our second condition is that its error is *bounded* in the norm $\|\cdot\|$ used to define our model. This ensures that the bound on $I(\hat{\omega})$ achieved via the optimization in Equation (19) implies a comparable

bound on the error of the derivative approximation $\langle \hat{\gamma}, h \rangle_n \approx d\psi(\hat{\lambda})(h)$ in Equation (18) for the relevant perturbation $h = \bar{\lambda} - \hat{\lambda}$.

**Assumption 3.2.** $\left\| \hat{\lambda}_u(\cdot, a) - \lambda_u(\cdot, a) \right\| = O_p(1)$ for all $u \le t$.

If our model is correctly specified in the sense that $\|\lambda_u\| < \infty$ for $\forall u$, a sensibly tuned $\|\cdot\|$-penalized estimator of $\lambda_u$ will have these two properties (see, e.g., Hirshberg & Wager, 2021, Remark 2).

Our third condition is that we have a sufficient degree of overlap.

**Assumption 3.3.** $\mathbb{E}[1/P(A = a, \widetilde{T} \ge u \mid X)] < \infty$

This is a substantially weakened version of the often-assumed 'strong overlap' condition that $P(A = a, \widetilde{T} \ge u \mid X)$ is bounded away from zero, allowing this probability to approach zero for some $X$ as long as it is 'typically' elsewhere.

Our final condition is, for the most part, a constraint on the complexity of our model.

**Assumption 3.4.** The unit ball $\mathcal{B} = \{h : \|h\| \le 1\}$ is Donsker and uniformly bounded in the sense that $\max_{h:\|h\| \le 1} \|h\|_\infty < \infty$. Furthermore, $\left\{ \frac{h(\cdot)}{P(A=a,\widetilde{T} \ge u | X = \cdot)} : \|h\| \le 1 \right\}$ is Donsker for all $u$.

The 'furthermore' clause here is implied by the first clause if strong overlap holds, as multiplication by the inverse probability weight $1/P(A = a, \widetilde{T} \ge u \mid X = \cdot)$ will not problematically increase the complexity of the set of functions $h \in \mathcal{B}$ *if those weights are bounded*.

We are now ready to state our main theoretical result, which involves the efficient influence function $\phi^{a,t}$ for estimating $\psi^{a,t}$,

$$
\begin{aligned}
\phi^{a,t}(O) &= S_t(X, a) - \psi^{a,t}(\lambda) + r(X, a)\mathbf{1}(A = a, G_u = 1)\omega_u(X_i, a)(Y_u - \lambda_u(X, a)) \\
&\text{for } r(X, a) = -S_t(X, a)\frac{S_{u-}(X, a)}{S_u(X, a)}
\end{aligned}
\tag{20}
$$

**Theorem 3.5.** Suppose $\hat{\lambda}$ is a hazard estimator fit on an auxiliary sample and Assumptions 3.1-3.4 are satisfied. Then the estimator $\hat{\psi}$ described in Equation (16), using the Riesz Representer estimate $\hat{\gamma}$ obtained by solving Equation (19) for any fixed $\sigma > 0$, is *asymptotically linear* with influence function $\phi$. That is, it has the asymptotic approximation $\hat{\psi} - \psi = \frac{1}{n} \sum_{i=1}^n \phi(O_i) + o_p(n^{-1/2})$.

It follows, via the central limit theorem, that under these conditions $\sqrt{n}(\hat{\psi} - \psi)$ is asymptotically normal with mean zero and variance $V = \mathbb{E}\left[\phi(\lambda)(O)^2\right]$. This justifies a standard approach to inference based on this asymptotic approximation, i.e., based on the t-statistic $\sqrt{n}(\hat{\psi} - \psi)/\hat{V}^{1/2}$ being approximately standard normal if $\hat{V}$ is a consistent estimator of this variance $V$.

As usual for estimators involving cross-fitting, working with multiple folds and averaging will yield an estimator with the same characterization without an auxiliary sample (e.g., Chernozhukov et al., 2018). The resulting estimator is asymptotically linear on the whole sample and, having the efficient influence function $\phi$, is asymptotically efficient.[2]

## 4 RELATED WORK

**Outcome regression and inverse probability weighting estimators.** Based on Proposition 2.2, the *outcome regression* approach (Makuch, 1982) estimates $\psi^{a,t}$ is by directly estimating the conditional event survival $S_t(a, X)$ given the treatment and covariates. Parametric and semi-parametric models such as the Cox proportional hazard (CoxPH) model (Cox, 1972; Andersen & Gill, 1982) in addition to deep learning techniques (Zhu et al., 2016; Katzman et al., 2018; Faraggi & Simon, 1995; Wang et al., 2021; Ching et al., 2018; Sun et al., 2020; Zhong et al., 2022; Nagpal et al., 2021;

---

[2]There are many equivalent formal descriptions of what asymptotic efficiency means (e.g.,in Van der Vaart, 2000, Chapter 25). In essence, it implies that no estimator can be reliably perform better in asymptotic terms, either in terms of criteria for point estimation like mean squared error or in terms of inferential behavior like the power of tests against local alternatives.

Meixide et al., 2022; Lv et al., 2022; Tibshirani, 1997; Biganzoli et al., 1998; Liestbl et al., 1994; Zhao & Feng, 2020; 2019; Hu et al., 2021; Luck et al., 2017; Yousefi et al., 2017; Lv et al., 2022) have been proposed to estimate $S_t(a, X)$. A popular alternative to outcome regression is the inverse probability of censoring and treatment weighting estimator (IPW) (Robins & Rotnitzky, 1992) defined as $1 - \frac{1}{n} \sum_{i=1}^{n} \frac{\mathbf{1}(A_i=a)}{\hat{\pi}(X_i,a)} \frac{\mathbf{1}(E_i=1)}{\hat{G}_{T_i}(X_i,a)} \mathbf{1}(\widetilde{T}_i \leq t)$. In the past, model misspecification is a problem for both approaches. Modern machine learning methods can readily alleviate this, but they are data hungry, which motivates the use of semi-parametric efficient estimators (Chernozhukov et al., 2017).

**Semi-parametric efficient estimators.**   The recent wake of big data and machine learning attracts renewed attention for the field of semi-parametric efficient estimation, with numerous notable contributions focusing on causal inference, such as Díaz (2020); Chernozhukov et al. (2017; 2022a); Robins & Rotnitzky (1992); Tsiatis (2006). One starts from the efficient influence function (EIF), which determines the fastest rate of any *regular* estimators (Van der Vaart, 2000), and construct efficient estimators e.g. the one-step estimator (Kennedy, 2022). Similar EIFs to 20 but with different parameterization are found in e.g. Díaz (2019); Cai & van der Laan (2020); Hubbard et al. (2000); Bai et al. (2013; 2017); Westling et al. (2023). All semi-parametric efficient estimators are the same asymptotically (Van der Vaart, 2000), whose asymptotic variances are determined by the EIF, similar to Theorem 3.5; this key property shows that our estimator is asymptotically as good as other semi-parametric efficient ones. It does not, however, characterize an estimator's finite-sample behavior.

**Covariate balancing.**   Numerous covariate balancing methods have been developed for estimating the effect of binary/continuous treatments on continuous outcomes including (Kallus & Santacatterina, 2021; 2022; Kallus, 2021; Hirshberg et al., 2019; Hirshberg & Wager, 2021; Zhao & Percival, 2017; Zhao et al., 2019; Wong & Chan, 2017; Visconti & Zubizarreta, 2018; Zubizarreta et al., 2014; Li et al., 2018; King et al., 2017; Josey et al., 2020; Yiu & Su, 2018; Hainmueller, 2012; Imai & Ratkovic, 2014; Zubizarreta, 2015, among others). There is however limited body of research on covariate balance in the context of survival data (Xue et al., 2023; Abraich et al., 2022; Leete et al., 2019; Santacatterina, 2023; Yiu & Su, 2022; Kallus & Santacatterina, 2021). Perhaps the most similar to our work and motivation is Xue et al. (2023), which is a covariate-balancing extension of Xie & Liu (2005). Unfortunately, they assume independent censoring ($T$ and $C$ are independent unconditionally of $X$) and their estimator is inefficient. None of the mentioned covariate-balancing methods in survival context provide results on asymptotic normality like ours, which allows for statistical inference, hypothesis testing, and the construction of confidence intervals.

## 5 EXPERIMENTS

We now describe the experimental evidence concerning our estimator. As noted by, e.g., Curth et al. (2021), ground truth is almost never available in causal inference, so synthetic or semi-synthetic data (i.e. synthesized data based on real data) is the standard. We focus on the experiments that provide the most insight into the behavior of our estimator and its competitors; additional experiments and metrics along with implementation details are included in Appendix A.

### 5.1 BASELINES, METRICS AND DATASETS

We compare our balancing estimator (**Balance**) with the outcome regression estimator (**OR**) and the doubly-robust estimator (**DR**). We include 2 implementation of **DR**, one with clipping the weight's denominator to be at least $10^{-3}$ and one without.

We report the average survival effect at time t: $\Delta^t = \psi^{1,t} - \psi^{0,t}$. Our key error metrics are (1) the relative root-mean-squared-error (**Relative RMSE**), defined as the RSME of an estimator divided by the RMSE of the **OR** estimator, which we take to be the baseline, and (2) the Bias over Standard Error ratio (**Bias/StdE**). Let $\{\hat{\Delta}_q^t\}_{q=1}^Q$ be our estimates over $Q$ simulations and $\Delta^t$ the ground truth, The RSME, Bias, and Standard Error are defined as $\sqrt{\frac{1}{Q} \sum_q (\hat{\Delta}_q^t - \Delta^t)^2}$, $\frac{1}{Q} \sum_q (\hat{\Delta}_q^t - \Delta^t)$, $\sqrt{\frac{1}{Q} \sum_q (\hat{\Delta}_q^t - \frac{1}{Q} \sum_q \hat{\Delta}_q^t)^2}$, respectively.

We use the two datasets from Curth et al. (2021) with slight modifications; note that they are concerned with the heterogeneous treatment effect and thus incomparable in our context. For the first dataset **Synthetic**, we modified the assignment distribution to $a \sim \text{Bern}(\text{Sigmoid}(\xi \times \sum_p x_p))$ where $x$ is a sample of a 10-dimensional multivariate normal, so that $\xi$ controls the lack of overlap from propensity (the higher $\xi$ the less overlap as Sigmoid saturates quicker). The default value of $\xi$ is 0.3. We refer readers to the authors' exposition of the different biases arising from censoring and assignment in this dataset. The second dataset **Twins** is a semi-synthetic dataset based on the Twins dataset (Louizos et al., 2017). For both, we set $t_{max} = 30$. We use 200 and 500 observations for **Synthetic** and **Twins**, respectively, for each $Q = 100$ simulations of each experiment. We provide the full data generating process in A.1, A.3.

## 5.2 DISCUSSION OF RESULTS

Poor overlap can effect estimators in 2 ways (1) make naive outcome regression estimators biased and (2) lead to extreme numerical inverses in inverse-weight-based estimators. We illustrate this with 2 experiments for censoring bias/poor overlap in time and treatment bias/poor overlap in propensity.

**Metrics over time.** From Figure 1, in both datasets, **Balance** is both accurate and inferentially useful i.e. we can construct meaningful confidence interval, without suffering from the poor overlap at higher times. **Balance** has comparable RMSE to **OR**, where **DR** and **DR-clip** are much worse; in **Twins**, **DR** clearly runs into large numerical inverses, and **DR-clip**, despite its improvements, is still far from **Balance**. Yet, **Bias/StdE** plots show that **OR** has serious bias issue that is exacerbated by censoring bias at higher times. This bias is extremely problematic for inference: an **Bias/StdE** of 1.4 at time $t = 15$ in **Synthetic** means that a nominal 95% confidence interval contains the truth only 74.5% of the time. In contrast, this number is always above 92% for **DR** and **Twins** (**Bias/StdE** $\leq 0.5$). Of course, as seen from **Relative RMSE** plots, **DR** and **DR-clip** have very high standard error that push their RMSE up and thus highly inaccurate.

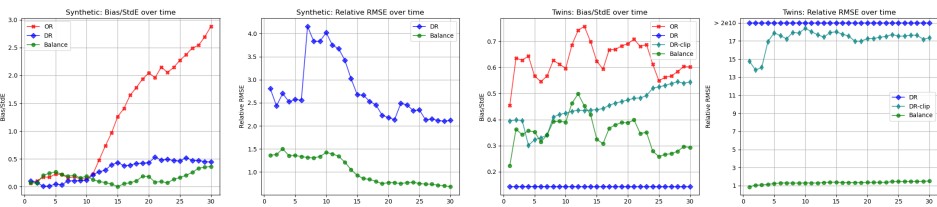

Figure 1: The effect of lack of overlap in propensity on **Bias/StdE** and **Relative RSME**.

**Metrics over varying propensity.** In Figure 2, we re-run **Synthetic** for $\xi \in \{0.1, 0.2, 0.3, 0.4, 0.5\}$ and report the metrics at $t = 15$ to see how poor overlap from extreme propensities influence our estimators. We observe similar behavior of estimators as seen in the first experiment. As overlap decreases, we see that both **OR**'s **Bias/StdE** and **DR**'s **Relative RMSE** increase. **DR-clip** was omitted in **Synthetic** plots as the estimates here are much less extreme and not truncated. Despite that, because of small sample sizes, we still see the same large errors in **DR**. In contrast, **Balance** is very stable across degrees of overlap, again proving its capability to fix **DR**'s accuracy problem while being less biased than **OR**, which was also shown in our semi-parametric efficiency theory.

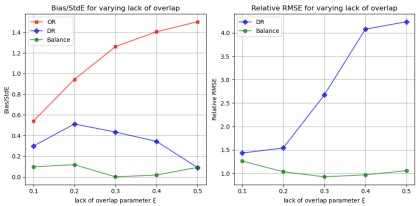

Figure 2: The effect of lack of overlap in time on **Bias/StdE** and **Relative RSME**.

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

# A    DESCRIPTION OF IMPLEMENTATION, DATASETS AND ADDITIONAL RESULTS

## A.1    IMPLEMENTATION

Throughout, we use the RBF kernel with length scale 10. For all methods considered, we need to train a hazard estimator for time-to-event (the survival function can then be constructed from the hazard). We use the discrete *logistic-hazard* model (Kvamme & Borgan, 2019) with the mean negative log-likelihood loss parameterized by the hazard function as:

$$L(\{O\}_{i=1}^n; \lambda) = -\frac{1}{n} \sum_{i=1}^n \sum_{u \le T_i} (Y_u \log \lambda_u(X_i, A_i) + (1 - Y_u) \log(1 - \lambda_u(X_i, A_i)))$$

where $Y_u = \mathbf{1}(E_i = 1, T_i = u)$. This loss breaks down into $|\mathcal{T}| \times 2$ independent binary cross-entropy losses for each $u \in \mathcal{T}$ and $a \in \{0, 1\}$:

$$L_u(\{O\}_{i=1}^n; \lambda) = \sum_{u \in \mathcal{T}} \sum_{a \in \{0,1\}} L_u(\{O\}_{i=1}^n; \lambda_u(\cdot, a)) \quad \text{where}$$

$$L_u(\{O\}_{i=1}^n; \lambda_u(\cdot, a)) = -\frac{1}{n} \sum_{i=1}^n \mathbf{1}(u \le T_i)(Y_u \log \lambda_u(X_i, A_i) + (1 - Y_u) \log(1 - \lambda_u(X_i, A_i)))$$

Therefore, we can fit $|\mathcal{T}| \times 2$ independent hazard models using kernel logistic regression.

**Covariate-balancing**    The hazard estimate must be fit on a separate split of the data, therefore we divide the data into 2 folds. For each fold, we fit the hazard estimator on the other fold, then estimate the Riesz representer using the just obtained hazard estimate on the canonical fold. The result is one causal parameter estimate of the canonical fold, by solving the optimization problem 19. This gives us the time-to-event hazard and the riesz estimates which we use to estimate the causal parameter of this canonical fold, using Equation (16). Lastly, we obtain the final estimate by averaging the estimates across 2 folds.

**Double robust estimation**    We need to train a hazard estimator for time-to-censoring, which can be done similarly to the time-to-event case (but with events flipped), and a propensity estimator, for which we use a simple linear logistic regression (correctly specified in both datasets). We use cross-fitting (Chernozhukov et al., 2018), where we randomly divide the dataset into K folds (we used K=5 in our experiments). For each fold, we fit the time-to-event/censoring hazard and propensity estimators on the remaining folds and obtain their estimates on the canonical fold. Using the time-to-censoring and the propensity estimates we obtain the riesz estimate using Equation (17). Lastly we obtain the causal parameter estimate using Equation (16).

## A.2    THE WEIGHT OPTIMIZATION PROBLEM

We start by observing that we do not need to solve the optimization Equation (19) all at once. It decomposes into $t$ separate optimizations over timestep-specific weights $\omega_{\cdot u}$ specific to a single timestep.

**Lemma A.1.** If the weights $\hat{\omega}$ solve Equation (19), the timestep-$u$-specific subset $\hat{\omega}_{\cdot u}$ satisfy the following one.

$$\hat{\omega}_u = \underset{\omega_u \in \mathbb{R}^n}{\arg\min} \left\{ I_u(\omega_u)^2 + \frac{\sigma^2}{n} \sum_{i=1}^n \hat{r}_u(X_i, a)^2 \mathbf{1}(A_i = a, G_{iu} = 1) \omega_{iu}^2 \right\} \quad \text{where}$$

$$I_u(\omega_u) = \max_{\|h_u\| \le 1} \frac{1}{n} \sum_{i=1}^n \left\{ \hat{r}_u(X_i, a) h_u(X_i) - \mathbf{1}(A_i = a, G_{iu} = 1) \hat{r}_u(X_i, a) \omega_{iu} h_u(X_i) \right\}$$

(21)

In the case that $\|\cdot\|$ is the norm of an RKHS, we can use the representer theorem to further simplify this optimization. The representer theorem implies that it is sufficient to maximize over $h_u$ that can

be written as $\sum_{j=1}^{n} \alpha_j k(\cdot, X_j)$ where $k$ is our space's kernel. Making this substitution, we get the following characterization in terms of the $n \times n$ kernel matrix $K$ satisfying $K_{ij} = k(X_i, X_j)$.

$$I_u(\omega_u) = \max_{\alpha^T K \alpha \leq 1} \left\{ \alpha^T K (\hat{r}_u \odot (1 - \mathcal{I}_u \odot \omega_u)) \right\}$$

where $\hat{r}_u, \mathcal{I}_u, \omega_u$ are vectors in $\mathbb{R}^n$ and $\hat{r}_{iu} = \hat{r}_u(X_i, a), \mathcal{I}_{iu} = \mathbf{1}(A_i = a, G_{iu} = 1)$, and $\odot$ is the element-wise product. The maximum is achieved at $\alpha = \hat{r}_u(1 - \mathcal{I}_u \odot \omega_u)/\|K^{1/2}\hat{r}_u \odot (1 - \mathcal{I}_u \odot \omega_u)\|$ and $I_u(\omega_u)$ is:

$$I_u(\omega_u) = \|K^{1/2}\hat{r}_u \odot (1 - \mathcal{I}_u \odot \omega_u)\|$$

Replacing into the outer minimization problem:

$$\hat{\omega}_u = \underset{\omega_u \in \mathbb{R}^n}{\arg\min} \left\{ (\hat{r}_u \odot (1 - \mathcal{I}_u \odot \omega_u))^T K \hat{r}_u \odot (1 - \mathcal{I}_u \odot \omega_u) + \frac{\sigma^2}{n} \sum_{i=1}^{n} \mathbf{1}(A_i = a, G_{iu} = 1)\hat{r}_{iu}^2 \omega_{iu}^2 \right\}$$

$$= \underset{\omega_u \in \mathbb{R}^n}{\arg\min} \left\{ (\hat{r}_u \odot \mathcal{I}_u \odot \omega_u)^T (K + \frac{\sigma^2}{n}I_n)(\hat{r}_u \odot \mathcal{I}_u \odot \omega_u) - 2\hat{r}_u^T K(\hat{r}_u \odot \mathcal{I}_u \odot \omega_u) \right\}$$

This quadratic-programming problem can be solved efficiently by most convex solvers, in particular we chose cvxpy. Now we show why we can decompose the original problem this way.

*Proof of theorem A.1.* Let $L_u(\omega_u, h_u)$ be defined as follows.

$$L_u(\omega_u, h_u) = \frac{1}{n}\sum_{i=1}^{n} \left\{ \hat{r}_{iu}h_u(X_i) - \mathbf{1}(A_i = a, G_{iu} = 1)\hat{r}_{iu}\omega_{iu}h_u(X_i) \right\}$$

In terms of this function,

$$I(\omega) \overset{def}{=} \max_{\sum_u \|h_u\|^2 \leq 1} \sum_u L_u(\omega_u, h_u)$$

$$= \max_{\substack{\beta \in \mathbb{R}^t \\ \sum_u \beta_u^2 \leq 1}} \sum_u \max_{\|h_u\| \leq \beta_u} L_u(\omega_u, h_u)$$

$$= \max_{\sum_u \beta_u^2 \leq 1} \sum_u \max_{\|h_u\| \leq 1} L_u(\omega_u, \beta_u h_u)$$

$$= \max_{\sum_u \beta_u^2 \leq 1} \sum_u \beta_u \max_{\|h_u\| \leq 1} L_u(\omega_u, h_u)$$

$$= \max_{\sum_u \beta_u^2 \leq 1} \sum_u \beta_u I_u(\omega_u)$$

$$= \sqrt{\sum_u I_u^2(\omega_u)}.$$

The last equality is due to Cauchy-Schwarz inequality, with equality achievable by setting $\beta_u$ proportional to $I_u(\omega_u)$. If we substitute this expression into our original optimization Equation (19), we get:

$$\hat{\omega} = \underset{\omega \in \mathbb{R}^{n|\mathcal{T}|}}{\arg\min} \left\{ \sum_{u \leq t} I_u(\omega_u)^2 + \frac{\sigma^2}{nt}\sum_{i=1}^{n}\sum_{u \leq t} \hat{r}_{iu}\mathbf{1}(A_i = a, G_{ui} = 1)\omega_{iu}^2 \right\}$$

$$= \underset{\omega \in \mathbb{R}^{n|\mathcal{T}|}}{\arg\min} \left\{ \sum_{u \leq t} \left( I_u(\omega_u)^2 + \frac{\sigma^2}{n}\sum_{i=1}^{n} \hat{r}_{iu}\mathbf{1}(A_i = a, G_{ui} = 1)\omega_{iu}^2 \right) \right\}$$

$$= \sum_{u \leq t} \underset{\omega \in \mathbb{R}^{n|\mathcal{T}|}}{\arg\min} \left( I_u(\omega_u)^2 + \frac{\sigma^2}{n}\sum_{i=1}^{n} \hat{r}_{iu}\mathbf{1}(A_i = a, G_{ui} = 1)\omega_{iu}^2 \right)$$

as each term in the sum over $u$ is a function of an individual $\omega_u$, we can therefore solve a separate minimization sub-problem for each $u$:

$$\hat{\omega}_u = \underset{\omega_u \in \mathbb{R}^n}{\arg\min} \left\{ I_u(\omega_u)^2 + \sigma^2 \sum_{i=1}^{n} \mathbf{1}(A_i = a, G_{iu} = 1)\hat{r}_{iu}^2 \omega_{iu}^2 \right\}$$

proving our claim.  □

A.3   DATASETS

We use both datasets from Curth et al. (2021) with modifications to the assignment distribution and censoring distribution to exacerbate the overlap problem.

**Synthetic Data.**

$$X \sim \mathcal{N}(0, 0.8 \times I_{10} + 0.2 \times J_{10})$$

$$A \sim \text{Bern}\left(0.2 \times \sigma\left(\sum_p x_p\right)\right)$$

$$h_t(a, x) = \begin{cases} 0.1\sigma(-5x_1^2 - a \times (\mathbf{1}(x_3 \geq 0) + 0.5)) & \text{for } t \leq 10 \\ 0.1\sigma(10x_2 - a \times (\mathbf{1}(x_3 \geq 0) + 0.5)) & \text{for } t > 10 \end{cases}$$

$$h_{C,t}(a, x) = \begin{cases} (0.01 \times \sigma(10x_4^2) & \text{for } t < 30 \\ 1 & \text{for } t \geq 30 \end{cases}$$

where $\sigma$ is a sigmoid function, $I_{10}$ is an identity matrix of size 10, $J_{10}$ is a $10 \times 10$ matrix of all ones, and $\sum_p x_p$ is the sum of all covariates of $x$. All time after $t = 30$ is censored so we set $t_{max} = 30$.

**Semi-synthetic Data.**   We preprocess the Twins dataset similar to Curth et al. (2021); Yoon et al. (2018). The time-to-event outcome is the time-to-mortality of each twin. Each observation $x$ has 30 covariates and we do not encode the categorical features. We are interested in the survival in the first 30 days, therefore $t_{max} = 30$. We also create artificial treatment and censoring:

$$A \sim \text{Bern}(\sigma(w_1^\top x + e)) \text{ where } w_1 \sim \text{Uniform}(-0.1, 0.1)^{30 \times 1} \text{ and } e \sim \mathcal{N}(0, 1^2)$$

$$C \sim \text{Exp}(10 \times \sigma(w_2^\top x)) \text{ where } w_2 \sim \mathcal{N}(0, 1^2)$$

treatment $A$ decides which twin outcome is observed. Here $C$ being continuous does not affect the discrete event time. We standardize covariate $x$ for training and only after creating the datasets.

A.4   ADDITIONAL RESULT FOR SYNTHETIC DATA.

We add 5 more metrics: MAE, MSE, Bias, Standard Error and Coverage for both datasets in figure 3 and 4. Overall, we see that **Balance** is competitive in all metrics. More specifically, it consistently has the lowest bias, its MAE and MSE are competitive to **OR** while not suffering from poor overlap at higher times in **Synthetic**. It also has consistently high coverage similar to **DR**, but does not suffer from high standard errors and therefore low accuracy. Both **DR** and **OR** suffer from poor overlap, but **DR** is the most susceptible, which shows that all the benefit of semi-parametric efficiency is lost to extreme inversions.

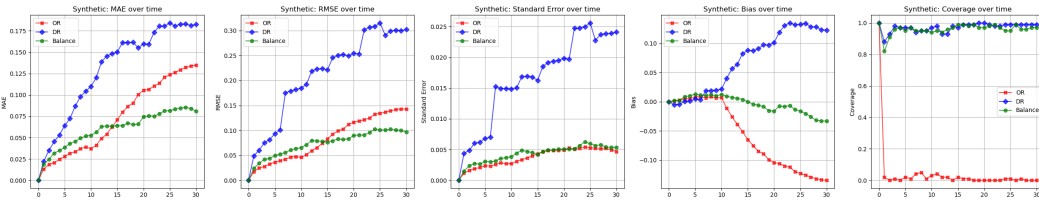

Figure 3: Additional metrics across time for **Synthetic**

In our experiment on varying degree of overlap in **Synthetic**, instead of a fixed time $t = 15$, we use 2 more metrics that summarize all time points: the Root Mean Squared Bias (**RISB**) and Root Mean Squared Error (**RISE**) (Xue et al., 2023), defined as $\sqrt{\frac{1}{T}\sum_u (\hat{\Delta}^t - \Delta^t)^2}$ and $\sqrt{\frac{1}{Q}\sum_q \frac{1}{T}\sum_u (\hat{\Delta}_q^t - \Delta^t)^2}$ respectively. Once again, we see that **Balance** is best among all methods across all degree of overlap (see figure 5).

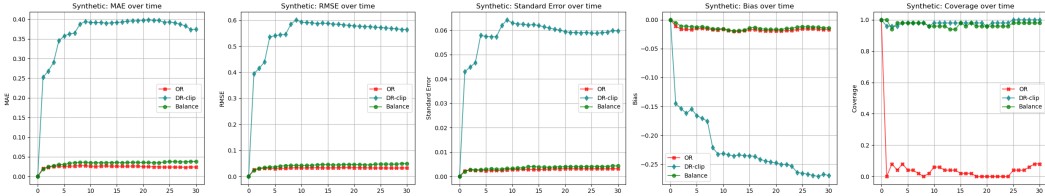

Figure 4: Additional metrics across time for **Twins**

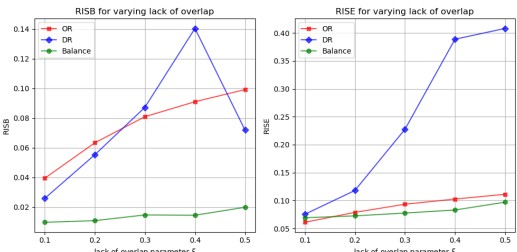

Figure 5: Effect of varying overlap on **RISB/RISE**

## B  IDENTIFICATION AND PROOF OF SUPPORTING LEMMAS

*Proof of Lemma 2.1.*

$$\lambda_t(x) = P(\widetilde{T} = t, E = 1|X = x)/P(\widetilde{T} \geq t|X = x)$$
$$= P(T = t, C \geq t|X = x)/P(\widetilde{T} \geq t|X = x)$$
$$= P(T = t|X = x)P(C \geq t|X = x)/P(\tilde{T} \geq t|X = x)$$
$$= \frac{P(T = t|X = x)}{P(T \geq t|X = x)} \frac{P(T \geq t|X = x)P(C \geq t|X = x)}{P(\widetilde{T} \geq t|X = x)}$$
$$= h_t(x)$$

and $H_t(x) = P(\widetilde{T} \geq t|X = x) = P(T \geq t, C \geq t|X = x) = P(T \geq t|X = x)P(C \geq t|X = x) = S_t(x)G_t(x)$ □

**Identification**   To identify the counterfactual survival function using the observable data, similar to Hubbard et al. (2000); Bai et al. (2013; 2017); Westling et al. (2023); Díaz (2019); Cai & van der Laan (2020), we require the following testable and untestable assumptions:

1. (A1) $T(a), C(a) \perp A|X$ for each $a \in \{0, 1\}$ ().
2. (A2) $T(a) \perp C(a)|A = a, X$ for each $a \in \{0, 1\}$.
3. (A3) $P(A = a|X) = 0 > 0$ almost surely.
4. (A4) $P(C(a) \geq \tau|X) > 0$ positivity (censoring),

in addition to consistency (A5) and non-interference (A6) Imbens & Rubin (2015). (A1), also referred to as no unmeasured confounders, selection on observables, exogeneity, and conditional independence, asserts that the potential outcomes/potential censoring times, and treatment assignment are independent given confounders. This assumption implies that all relevant information regarding treatment assignment and follow-up censoring times is captured in the available data. (A2) states that the potential follow-up and censoring times are independent given the treatment and confounders. (A3) and (A4) assert that the probability of receiving or not receiving treatment, as well as the probability of censoring, given confounders, is greater than zero. Finally, (A6) and (A7) state that the observed treatment corresponds to the actual treatment received, ensuring consistency between the observed and true treatment assignments.

*Proof of Proposition 2.2.*

$$
\begin{aligned}
P(T(a) > t) &= \mathbb{E}[P(T(a) > t)|X] \\
&= \mathbb{E}[P(T > t|X, A = a)] \\
&= \mathbb{E}[S_t(X, a)] \\
&= \mathbb{E}\left[\prod_{u \leq t}(1 - h_t(X, a))\right] \\
&= \mathbb{E}\left[\prod_{u \leq t}(1 - \lambda_t(X, a))\right]
\end{aligned}
$$

where we used iterated expectation in the first equation, (A1) in the second equation, (A2) in the forth equation, and Lemma 2.1 in the last equation. $\qquad\square$

## C  PROOF OF THEOREM 3.5

We first recall and introduce additional notations for this section. Denote $Z = (X, A, G)$ the conditioning structure of the hazard function $\lambda$. As $r$ and $S$ are functions of $\lambda$, we use $\hat{r}$ and $\hat{S}$ as functions of $\hat{\lambda}$ to denote the estimator counterparts. We drop the superscripts $a, t$ of $\psi^{a,t}$ since they are not relevant in the proof. We use $\psi = \psi(\lambda)$ for the parameter of interest $\mathbb{E}[S_t(X, a)]$, $\psi_n(\lambda)$ for the sample analog $\sum_{i=1}^n S_t(X, a)$. As $S$ and $\lambda$ are one-to-one, $\psi(\hat{\lambda})$ would be $\mathbb{E}[\hat{S}_t(X, a)]$. We use $\hat{\psi}$ to denote our estimator, i.e.,

$$\hat{\psi} = \psi_n(\hat{\lambda}) + \frac{1}{n}\sum_{i=1}^n\sum_{u \leq t} \hat{\gamma}_{iu}(\mathbf{1}(E_i = 1, \widetilde{T} = u) - \hat{\lambda}_u(Z_i)) \tag{22}$$

Let's start by recalling what we are proving.

$$\hat{\psi} - \psi = \frac{1}{n}\sum_{i=1}^n \phi_\lambda(O_i) + o_p(n^{-1/2}) \quad \text{where} \tag{23}$$

$$\phi(\lambda)(O) = (S_t(X, a) - \psi)$$
$$+ \sum_{u \leq t} r_u(X, a)\mathbf{1}(A = a, G_u = 1)\omega_u(X, a)\left(\mathbf{1}(E = 1, \widetilde{T} = u) - \lambda_u(X, a)\right),$$
$$\omega_u(X, a) = \frac{1}{H_{u-}(X, a)\pi(X, a)},$$
$$r_u(X, a) = -S_t(X, a)\frac{S_{u-}(X, a)}{S_u(X, a)}.$$

To do this, we will work with this error decomposition.

$$\hat{\psi} - \psi(\lambda) = \psi_n(\hat{\lambda}) + \frac{1}{n}\sum_{i=1}^n\sum_{u \leq t}\hat{\gamma}_{iu}\{Y_{iu} - \hat{\lambda}_u(Z_i)\} - \psi(\lambda)$$
$$= \left\{\psi_n(\hat{\lambda}) - \psi(\hat{\lambda})\right\}$$
$$+ \frac{1}{n}\sum_{i=1}^n\sum_{u \leq t}\hat{\gamma}_{iu}(Y_{iu} - \hat{\lambda}_u(Z_i)) - d\psi(\hat{\lambda})(\lambda - \hat{\lambda}) \tag{24}$$
$$+ \left\{\psi(\hat{\lambda}) + d\psi(\hat{\lambda})(\lambda - \hat{\lambda}) - \psi(\lambda)\right\}.$$

We prove Equation (23) in three steps. Throughout, we will work conditionally on the auxilliary sample used to estimate $\hat{\lambda}$, so we can act as if it is a deterministic function. This will imply that our claim holds where $o_p$ refers to probability conditional on the auxilliary sample and therefore also that it holds where $o_p$ refers to unconditional probability.

**Step 1.**  The third term in this decomposition, the error of our linearization of $\psi$ around $\hat{\lambda}$, is $o_p(n^{-1/2})$. Lemma C.2 below shows that this is implied by our assumption that $\hat{\lambda}$ converges $\lambda$ at faster-than-fourth-root rate.

**Step 2.**  Lemma C.4 below concludes that the second term in our decomposition has the following asymptotic approximation.

$$\frac{1}{n}\sum_{i=1}^n\sum_{u \leq t}\hat{r}_u(X, a)\mathbf{1}(A_i = a, G_u = 1)\omega_u(X, a)\{\mathbf{1}(E_i = 1, \widetilde{T}_i = u) - \lambda_u(Z_i)\} + o_p(n^{-1/2})$$

**Step 3.** The sum of the first term in our decomposition and the non-negligible part of the second is

$$= \psi_n(\hat{\lambda}) - \psi(\hat{\lambda}) + \frac{1}{n} \sum_{i=1}^{n} \sum_{u \le t} \hat{r}(X_i, a) \mathbf{1}(A_i = a, G_{iu} = 1) \omega_u(X_i, a) \{ \mathbf{1}(E_i = 1, \widetilde{T}_i = u) - \lambda_u(Z_i) \}$$

$$\stackrel{def}{=} \frac{1}{n} \sum_{i=1}^{n} \hat{\phi}(O_i)$$

(25)

To complete our proof of the claim Equation (23), we show

$$\frac{1}{n} \sum_{i=1}^{n} \left\{ \hat{\phi}(O_i) - \phi(O_i) \right\} = o_p(n^{1/2}).$$

Because this is an average of independent and identically distributed terms with mean zero, its mean square is $1/n$ times the variance of an individual term; thus, all we have to do is show that the variance of $(\hat{\phi} - \phi)(O_i)$ goes to zero. In Lemma C.5 below, we show that this is a consequence of the convergence of $\hat{\lambda} \to \lambda$.

We conclude by stating and proving our lemmas.

**Lemma C.1.** For all $t \in \mathcal{T}$,

$$\hat{S}_t(X, a) - S_t(X, a) = \sum_{u \le t} \hat{S}_t(X, a) \frac{S_{u-}(X, a)}{\hat{S}_{u-}(X, a)} \frac{\hat{S}_{u-}(X, a)}{\hat{S}_u(X, a)} \left( \lambda_u(X, a) - \hat{\lambda}_u(X, a) \right).$$

Furthermore,

$$\left\| S_t(X, a) - \hat{S}_t(X, a) \right\|_{L_2(P)} \le |\mathcal{T}| \max_{u \le t} \left\| \lambda_u(X, a) - \hat{\lambda}_u(X, a) \right\|_{L_2(P)}$$

**Lemma C.2.** Let $\hat{\lambda}$ and $\lambda$ be two hazards and $\hat{S}$ and $S$ the associated survival functions. Then the functional $\psi(h)$ evaluated at $h = \lambda$ has the following expansion:

$$\psi(\lambda) = \psi(\hat{\lambda}) + \mathbb{E}\left[ -\sum_{u \le t} \hat{S}_t(X, a) \frac{\hat{S}_{u-}(X, a)}{\hat{S}_u(X, a)} \left( \lambda_u(X, a) - \hat{\lambda}_u(X, a) \right) \right]$$

$$+ \mathbb{E}\left[ -\sum_{u \le t} \frac{\hat{S}_t(X, a)}{\hat{S}_u(X, a)} \left( S_{u-}(X, a) - \hat{S}_{u-}(X, a) \right) \left( \lambda_u(X, a) - \hat{\lambda}_u(X, a) \right) \right]$$

(26)

Furthermore, under Assumption 3.1 and Lemma C.1, the second term is $o_p(n^{-1/2})$, therefore:

$$\psi(\hat{\lambda}) = \psi(\lambda) + \mathbb{E}\left[ \sum_{u \le t} \hat{S}_t(X, a) \frac{\hat{S}_{u-}(X, a)}{\hat{S}_u(X, a)} \left( \lambda_u(X, a) - \hat{\lambda}_u(X, a) \right) \right] + o_p(n^{-1/2})$$

**Remark C.3.** Since the second term in the expansion of $\psi(\lambda)$, shown in Equation (26), is linear in $(\lambda - \hat{\lambda})$ and the third is higher order, this lemma shows that the second term is indeed the derivative $d\psi(\hat{\lambda})(\lambda - \hat{\lambda})$ that appears in Equation (11).

**Lemma C.4.** Suppose the assumptions of Theorem 3.5 are satisfied.

$$\frac{1}{n} \sum_{i=1}^{n} \sum_{u \le t} \hat{\gamma}_{iu} \{ \mathbf{1}(E_i = 1, \widetilde{T}_i = u) - \hat{\lambda}_u(Z_i) \} - d\psi(\bar{\lambda} - \hat{\lambda})(\lambda)$$

$$= \frac{1}{n} \sum_{i=1}^{n} \sum_{u \le t} \hat{r}_u(X, a) \mathbf{1}(A_i = a, G_u = 1) \omega_u(X, a) \{ \mathbf{1}(E_i = 1, \widetilde{T}_i = u) - \lambda_u(Z_i) \} + o_p(n^{-1/2}).$$

This remains true if Assumption 3.1's $o_p(n^{-1/4})$ rate assumption is weakened to an $o_p(1)$ rate.

**Lemma C.5.** Suppose our overlap assumption, Assumption 3.3, is satisfied. Then the influence function $\phi$ is mean-square continuous as a function of $\lambda$, i.e.,

$$\phi(\lambda)(O_i) = \{S_t(X_i, a) - \mathbb{E}[S_t(X, a)]\}$$
$$+ \sum_{u \leq t} r_u(X_i, a)\mathbf{1}(A_i = a, G_{iu} = 1)\omega_u(X_i, a)\left(\mathbf{1}(\widetilde{T}_i = u, E_i = 1) - \lambda_u(X_i, a)\right)$$

satisfies $\mathbb{E}\{\phi(\hat{\lambda}) - \phi(\lambda)\}^2 \to 0$ if $\hat{\lambda}$ and $\lambda$ are two hazards, with corresponding survival curves $\hat{S}$ and $S$ and ratios $\hat{r}$ and $r$, that converge in the sense that $\left\|\hat{\lambda} - \lambda\right\|_{L_2(P)} \to 0$.

## C.1 LEMMA PROOFS

*Proof of Lemma C.1.* For all $t \in \mathcal{T}$

$$\hat{S}_t(X, a) - S_t(X, a) = \hat{S}_t(X, a)\left(1 - \frac{S_t(X, a)}{\hat{S}_t(X, a)}\right)$$
$$= \hat{S}_t(X, a)\sum_{u \leq t}\left(\frac{S_{u-}(X, a)}{\hat{S}_{u-}(X, a)} - \frac{S_u(X, a)}{\hat{S}_u(X, a)}\right)$$
$$= \hat{S}_t(X, a)\sum_{u \leq t}\frac{S_{u-}(X, a)}{\hat{S}_{u-}(X, a)}\frac{\hat{S}_{u-}(X, a)}{\hat{S}_u(X, a)}\left(\frac{\hat{S}_u(X, a)}{\hat{S}_{u-}(X, a)} - \frac{S_u(X, a)}{S_{u-}(X, a)}\right)$$
$$= \hat{S}_t(X, a)\sum_{u \leq t}\frac{S_{u-}(X, a)}{\hat{S}_{u-}(X, a)}\frac{\hat{S}_{u-}(X, a)}{\hat{S}_u(X, a)}\left((1 - \hat{\lambda}_u(X, a)) - (1 - \lambda_u(X, a))\right)$$
$$= \sum_{u \leq t}\hat{S}_t(X, a)\frac{S_{u-}(X, a)}{\hat{S}_{u-}(X, a)}\frac{\hat{S}_{u-}(X, a)}{\hat{S}_u(X, a)}\left(\lambda_u(X, a) - \hat{\lambda}_u(X, a)\right)$$

$$(27)$$

therefore

$$\left\|S_t(X, a) - \hat{S}_t(X, a)\right\|_{L_2(P)}$$
$$= \left\|\sum_{u \leq t}\hat{S}_t(X, a)\frac{S_{u-}(X, a)}{\hat{S}_{u-}(X, a)}\frac{\hat{S}_{u-}(X, a)}{\hat{S}_u(X, a)}\left(\lambda_u(X, a) - \hat{\lambda}_u(X, a)\right)\right\|_{L_2(P)}$$
$$\leq |\mathcal{T}|\max_{u \leq t}\left\|\hat{S}_t(X, a)\frac{S_{u-}(X, a)}{\hat{S}_{u-}(X, a)}\frac{\hat{S}_{u-}(X, a)}{\hat{S}_u(X, a)}\left(\lambda_u(X, a) - \hat{\lambda}_u(X, a)\right)\right\|_{L_2(P)}$$
$$\leq |\mathcal{T}|\max_{u \leq t}\left\|\hat{S}_t(X, a)\frac{S_{u-}(X, a)}{\hat{S}_u(X, a)}\left(\lambda_u(X, a) - \hat{\lambda}_u(X, a)\right)\right\|_{L_2(P)}$$
$$\leq |\mathcal{T}|\max_{u \leq t}\left\|\lambda_u(X, a) - \hat{\lambda}_u(X, a)\right\|_{L_2(P)}$$

since $0 \leq \frac{\hat{S}_t(X,a)S_{u-}(X,a)}{\hat{S}_u(X,a)} \leq \frac{\hat{S}_t(X,a)}{\hat{S}_u(X,a)} \leq 1$. □

*Proof of Lemma C.2.* We expand each term of the decomposition in Lemma C.1 around the approximation $S_{u-}(X,a)/\hat{S}_{u-}(X,a) \approx 1$,

$$\hat{S}_t(X,a)\frac{S_{u-}(X,a)}{\hat{S}_{u-}(X,a)}\frac{\hat{S}_{u-}(X,a)}{\hat{S}_u(X,a)}\left(\lambda_u(X,a) - \hat{\lambda}_u(X,a)\right)$$

$$= \hat{S}_t(X,a)\frac{\hat{S}_{u-}(X,a)}{\hat{S}_u(X,a)}\left(\lambda_u(X,a) - \hat{\lambda}_u(X,a)\right)$$

$$+ \hat{S}_t(X,a)\left(\frac{S_{u-}(X,a)}{\hat{S}_{u-}(X,a)} - 1\right)\frac{\hat{S}_{u-}(X,a)}{\hat{S}_u(X,a)}\left(\lambda_u(X,a) - \hat{\lambda}_u(X,a)\right)$$

$$= \hat{S}_t(X,a)\frac{\hat{S}_{u-}(X,a)}{\hat{S}_u(X,a)}\left(\lambda_u(X,a) - \hat{\lambda}_u(X,a)\right)$$

$$+ \frac{\hat{S}_t(X,a)}{\hat{S}_u(X,a)}\left(S_{u-}(X,a) - \hat{S}_{u-}(X,a)\right)\left(\lambda_u(X,a) - \hat{\lambda}_u(X,a)\right)$$

therefore

$$\psi(\hat{\lambda}) - \psi(\lambda) = \mathbb{E}\left[\hat{S}_t(X,a) - S_t(X,a)\right]$$

$$= \mathbb{E}\left[\sum_{u\leq t}\hat{S}_t(X,a)\frac{S_{u-}(X,a)}{\hat{S}_{u-}(X,a)}\frac{\hat{S}_{u-}(X,a)}{\hat{S}_u(X,a)}\left(\lambda_u(X,a) - \hat{\lambda}_u(X,a)\right)\right] \quad \text{(By Lemma C.1)}$$

$$= \mathbb{E}\left[\sum_{u\leq t}\hat{S}_t(X,a)\frac{\hat{S}_{u-}(X,a)}{\hat{S}_u(X,a)}\left(\lambda_u(X,a) - \hat{\lambda}_u(X,a)\right)\right]$$

$$+ \mathbb{E}\left[\sum_{u\leq t}\frac{\hat{S}_t(X,a)}{\hat{S}_u(X,a)}\left(S_{u-}(X,a) - \hat{S}_{u-}(X,a)\right)\left(\lambda_u(X,a) - \hat{\lambda}_u(X,a)\right)\right]$$

$$\tag{28}$$

We now argue that the second term is $o(n^{-1/2})$:

$$\left|\mathbb{E}\left[\sum_{u\leq t}\frac{\hat{S}_t(X,a)}{\hat{S}_u(X,a)}\left(S_{u-}(X,a) - \hat{S}_{u-}(X,a)\right)\left(\lambda_u(X,a) - \hat{\lambda}_u(X,a)\right)\right]\right|$$

$$\leq \left\|\sum_{u\leq t}\frac{\hat{S}_t(X,a)}{\hat{S}_u(X,a)}\left(S_{u-}(X,a) - \hat{S}_{u-}(X,a)\right)\left(\lambda_u(X,a) - \hat{\lambda}_u(X,a)\right)\right\|_{L_2(P)}$$

$$\leq |\mathcal{T}|\max_{u\leq t}\left\|\left(S_{u-}(X,a) - \hat{S}_{u-}(X,a)\right)\left(\lambda_u(X,a) - \hat{\lambda}_u(X,a)\right)\right\|_{L_2(P)}$$

$$\leq |\mathcal{T}|\max_{u\leq t}\left\|S_{u-}(X,a) - \hat{S}_{u-}(X,a)\right\|_{L_2(P)}\left\|\lambda_u(X,a) - \hat{\lambda}_u(X,a)\right\|_{L_2(P)}$$

We then use the bound for $S$ in Lemma C.1 to get:

$$\left|\mathbb{E}\left[\sum_{u\leq t}\frac{\hat{S}_t(X,a)}{\hat{S}_u(X,a)}\left(S_{u-}(X,a) - \hat{S}_{u-}(X,a)\right)\left(\lambda_u(X,a) - \hat{\lambda}_u(X,a)\right)\right]\right|$$

$$\leq |\mathcal{T}|^2\max_{u\leq t}\left\|\lambda_u(X,a) - \hat{\lambda}_u(X,a)\right\|_{L_2(P)}^2$$

$$= o_p(n^{-1/2}) \qquad \text{(By Assumption 3.1)}$$

$\square$

*Proof of Lemma C.4.* We will establish this result timestep-by-timestep. That is, we will show that for all $u$,

$$
\frac{1}{n} \sum_{i=1}^{n} \hat{\gamma}_{iu} \{ Y_{iu} - \hat{\lambda}_u(X_i, A_i, G_i) \} - \mathbb{E}[\hat{r}_u(X, a)(\bar{\lambda} - \hat{\lambda})(X, a, 1)]
$$
$$
= \frac{1}{n} \sum_{i=1}^{n} \gamma_u(X_i, A_i, G_i) \{ Y_{iu} - \bar{\lambda}(X_i, A_i, G_i) \} + o_p(n^{-1/2}). \tag{29}
$$

It's sufficient to show that this holds for a version in which the expectation is replaced by a sample average, as the variance of the difference is $o(1/n)$. This follows from the consistency of $\hat{\lambda}$ and the boundedness of $\hat{r}_u(\cdot, a)$.[3]

$$
\mathrm{Var} \left[ \frac{1}{n} \sum_{i=1}^{n} \hat{r}_u(X_i, a)(\bar{\lambda}_u - \hat{\lambda}_u)(X_i, a, 1) - \mathbb{E} r_u(X_i, a)(\bar{\lambda} - \hat{\lambda})(X_i, a, 1) \right]
$$
$$
\leq \frac{\mathbb{E} \left[ \hat{r}_u(X_i, a)^2 (\bar{\lambda}_u - \hat{\lambda}_u)(X_i, a, 1)^2 \right]}{n}
$$
$$
\leq \frac{\| r_u(\cdot, a) \|_\infty \left\| \bar{\lambda}_u(\cdot, a, 1) - \hat{\lambda}_u(\cdot, a, 1) \right\|_{L_2(P)}}{n}.
$$

To do this, we will start with the result of Lemma A.1, which characterizes the weights $\hat{\gamma}_{u\cdot}$ as the solution to an optimization problem equivalent to the following one.

$$
\hat{\gamma}_u = \underset{\gamma_u \in \mathbb{R}^n}{\arg\min} \left\{ I_u(\omega_u)^2 + \frac{\sigma^2}{n} \sum_{i=1}^{n} \gamma_u^2 \right\} \quad \text{where}
$$
$$
I_u(\omega) = \max_{\substack{\|f(\cdot, a, g)\| \leq 1 \\ \forall a, g}} \frac{1}{n} \sum_{i=1}^{n} h(X_i, A_i, G_{iu}, f) - \gamma_{iu} f(X_i, A_i, G_{iu}) \tag{30}
$$
$$
\text{for} \quad h(X, A, G, f) = \hat{r}(X, a) f(X, a, 1)
$$

This differs from the characterization from Lemma A.1 in two ways. First, we do not restrict the parametric form of the weights to be $\gamma_{iu} = \mathbf{1}(A_i = a, G_{iu} = 1) G_{iu} \hat{r}(X_i, a) \omega_i$. Second, we write the class of functions we're maximizing over as functions of $(X, A, G)$ instead of functions of $X$ alone. However, as argued in Hirshberg et al. (2019, Proposition 7), the solution must satisfy $\hat{\gamma}_{iu} = 0$ unless $A_i = a$ and $G_{iu} = 1$, as to do otherwise would increase the objective function. Thus, we may impose this restriction on $\gamma_{iu}$ and therefore reduce our maximization to one over $f(X, a, 1)$ without changing the solution $\hat{\gamma}_u$. Reparameterizing in terms of $\omega_{iu} = \gamma_{iu}/\hat{r}(X, a)$ yields the equivalent problem described in Lemma A.1. The benefit of the formulation we use here is that it's recognizable as an instance of the problem used to estimate weights in Hirshberg & Wager (2021).

Under our assumptions, Hirshberg & Wager (2021, Theorem 1) establishes this claim, i.e. the sample-average-replacing-expectation version of Equation (29). Almost. It does so for a *fixed* function $\hat{r}_u$; because ours varies with sample size we would need a triangular-array version. However, such a version follows from the proof used to derive Hirshberg & Wager (2021, Theorem 1) from a finite sample result (Hirshberg & Wager, 2021, Theorem 2). In particular, the finite sample result shows that the approximation we've claimed holds with error that's bounded in terms of certain Rademacher complexity fixed points, and the proof of Hirshberg & Wager (2021, Theorem 1) shows that the Donsker conditions assumed there ensure that these bounds are $o_p(n^{-1/2})$. Following the same argument, the Donsker conditions we've assumed here, in conjunction with the uniform-in-n boundedness of $\hat{r}_u(\cdot, a)$ and the contraction principle for Rademacher complexity, imply the same. □

---

[3] We can treat $\hat{\lambda} - \bar{\lambda}$ as deterministic here, avoiding empirical process arguments, because $\hat{\lambda}$ is cross-fit.

*Proof of Lemma C.5.*

$$\phi(\hat{\lambda})(O_i) - \phi(\lambda)(O_i)$$

$$= \left( \hat{S}_t(X_i, a) - E\left[ \hat{S}_t(X, a) \right] \right)$$

$$+ \sum_{u \leq t} \hat{r}_u(X_i, a) \mathbf{1}(A_i = a, G_{iu} = 1) \omega_u(X_i, a) \left( \mathbf{1}(\widetilde{T}_i = u, E_i = 1) - \lambda_u(X_i, a) \right)$$

$$- (S_t(X_i, a) - E\left[ S_t(X, a) \right])$$

$$- \sum_{u \leq t} r_u(X_i, a) \mathbf{1}(A_i = a, G_{iu} = 1) \omega_u(X_i, a) \left( \mathbf{1}(\widetilde{T}_i = u, E_i = 1) - \lambda_u(X_i, a) \right)$$

$$= (\hat{S}_t(X_i, a) - S_t(X_i, a)) - \left( E\left[ \hat{S}_t(X, a) \right] - E\left[ S_t(X, a) \right] \right)$$

$$+ \sum_{u \leq t} (\hat{r}_u(X_i, a) - r_u(X_i, a)) \mathbf{1}(A_i = a, G_{iu} = 1) \omega_u(X_i, a) \left( \mathbf{1}(\widetilde{T}_i = u, E_i = 1) - \lambda_u(X_i, a) \right)$$

For simplicity, we drop the $(X_i, a)$ when writing functions $\hat{S}, S, \hat{r}_u, r_u, \hat{\lambda}_u,$ and $\lambda_u$.

We first consider the term $\hat{S}_t - S_t$. From Lemma C.1, we can write

$$\hat{S}_t - S_t = \sum_{u \leq t} \hat{S}_t \frac{S_{u-}}{\hat{S}_{u-}} \frac{\hat{S}_{u-}}{\hat{S}_u} \left( \lambda_u - \hat{\lambda}_u \right) = \sum_{u \leq t} \hat{S}_t \frac{S_{u-}}{\hat{S}_u} \left( \lambda_u - \hat{\lambda}_u \right)$$

It is obvious that $0 \leq \frac{\hat{S}_t S_{u-}}{\hat{S}_u} \leq \frac{\hat{S}_t}{\hat{S}_u} \leq 1$. Applying Holder's inequality and the given condition $\left\| \hat{\lambda} - \lambda \right\|_{L_2(P)} \to 0$, we can imply that $\hat{S} \to S$.

Since $\hat{r}_u$ and $r_u$ are continuous functions of $\hat{S}$ and $S$, respectively, and $\hat{S} \to S$ as shown above, $\hat{r}_u \to r_u$. Then for the last term of the $\phi(\hat{\lambda})(O_i) - \phi(\lambda)(O_i)$, we can continue applying Holder's inequality and further conclude that $\mathbb{E}\{\phi(\hat{\lambda}) - \phi(\lambda)\}^2 \to 0$. □

## C.2 A Sketch of Theorem 3.5

Our proof of Theorem 3.5 uses results from Hirshberg & Wager (2021) to do some heavy lifting. For the sake of self-containedness, we will sketch the main ideas of the argument we'd use to prove it from scratch. We use a more detailed decomposition of the error $\hat{\psi} - \psi(\lambda)$ as follows:

$$\hat{\psi} - \psi(\lambda) = \psi_n(\hat{\lambda}) + \frac{1}{n} \sum_{i=1}^{n} \sum_{u \leq t} \hat{\gamma}_{iu} \{ Y_{iu} - \hat{\lambda}_u(Z_i) \} - \psi(\lambda)$$

$$= \left\{ \psi_n(\hat{\lambda}) - \psi(\hat{\lambda}) \right\} + \frac{1}{n} \sum_{i=1}^{n} \sum_{u \leq t} \hat{\gamma}_{iu}(Y_{iu} - \lambda(Z_i))$$

$$+ \frac{1}{n} \sum_{i=1}^{n} \sum_{u \leq t} \hat{\gamma}_{iu}(\lambda_{iu} - \hat{\lambda}_u(Z_i)) - \frac{1}{n} \sum_{i=1}^{n} \sum_{u \leq t} \hat{r}_u(X_i, a)(\lambda(Z_i) - \hat{\lambda}(Z_i)) \quad (31)$$

$$+ \frac{1}{n} \sum_{i=1}^{n} \sum_{u \leq t} \hat{r}_u(X_i, a)(\lambda(Z_i) - \hat{\lambda}(Z_i)) - d\psi(\hat{\lambda})(\lambda - \hat{\lambda})$$

$$+ \left\{ \psi(\hat{\lambda}) + d\psi(\hat{\lambda})(\lambda - \hat{\lambda}) - \psi(\lambda) \right\}.$$

We sketch the analysis of each of the 4 terms above:

1. The first term converges to the influence function of the estimator because $\hat{\lambda}$ and $\hat{\gamma}$ are convergent. That the latter converges to the population Riesz representer is a consequence of the analysis of the imbalance in the 2nd term below.

2. The 2nd term is the imbalance term motivated by the approximation 18, and our optimization problem 19 directly controls it. This is exactly where we borrow the covariate-balancing analysis of Hirshberg & Wager (2021) to our problem, noting that they have similar structure.

3. The 3rd term is the difference of the sample-average derivative and its expectation, can be shown to be $o(n^{-1/2})$ because each term of the mean $(\sum_{u \leq t} \hat{r}_u(X_i, a)(\lambda(Z_i) - \hat{\lambda}(Z_i)) - d\psi(\hat{\lambda})(\lambda - \hat{\lambda}))$ has mean 0 and variance $o(1)$ as consequence of the convergence of $\hat{\lambda}$.

4. The 4th term is the 2nd-order remainder as before and is $o(n^{-1/2})$.

Overall, we see again that $\hat{\psi} - \psi(\lambda) = \sum_{i=1}^{n} \phi(O_i) + o(n^{1/2})$.

