# OpenReview forum: "STABLE ESTIMATION OF SURVIVAL CAUSAL EFFECTS"
_ICLR.cc/2024/Conference — Submitted to ICLR 2024_

### Official Review · Reviewer_CT2T · 2023-10-26

**Soundness:** 3 good
**Presentation:** 2 fair
**Contribution:** 2 fair
**Rating:** 5
**Confidence:** 4

**Summary:**

The authors explored the causal survival effects problem. They contended that conventional approaches relying on inverse weighting were prone to instability, where minor inaccuracies in weight estimation could result in substantial errors in the overall estimation. To enhance their estimations, they explored the expansion of the survival function via Taylor's expansion and leveraged the Reisz representation theorem for derivative estimation.They also demonstrated that their method exhibits asymptotic efficiency under a specific set of conditions.

**Strengths:**

The utilization of the Reisz representation theorem for estimating the derivative of the survival function is a clever innovation. Their definition of the Reisz representer in the context of inverse probability weights serves to circumvent the issues encountered in methodologies reliant on inverse weighting.

**Weaknesses:**

- The proposed method appears to operate on a population level, yet all steps involve conditioning the hazard function on covariates. I expected to see individual-level results, and it's unclear how well the method performs at the individual level.
- In section 2.2 (identification), there appears to be a typo in A3.
- It's not evident which distribution the expectations are calculated on in the text.
- The natural proxy used for hazard estimation can be quite noisy, especially in small datasets.
- In section 2.3, there is a change in notation from \lambda(x, a) to \lambda(a, x). Consistent notation would be preferable.
- The proof of Lemma 2.1 requires a correction; the survival function should be denoted as P(T > t).

**Questions:**

- Could you please explain the significance of the second metric, bias/stde?
- I noticed that in both the synthetic experiments (prior to t=15) and in the twins dataset, your bias exceeds the odds ratio (OR). What accounts for this discrepancy?
- If the odds ratio (OR) is already displaying high bias, why did you introduce the concept of relative RMSE? Why not employ RMSE and compare it to the OR in addition?
-In the section on covariate balancing, the paper references Xue et al. 2023, which assumes independent censoring. I believe your method makes a similar assumption. If this is not the case, could you please clarify?

---

> ### Author Response · Authors · 2023-11-23
> **Thank you for your insightful comments and suggestions. We've taken your feedback into careful consideration and made the following clarifications and improvements.**
>
> Thank you for your insightful comments and suggestions. We've taken your feedback into careful consideration and made the following clarifications and improvements:
>
> 1. Population vs Individual Level: Our method addresses the average treatment effect (ATE) rather than individual or heterogeneous treatment effects (HTE).  Conditioning on the covariate is necessary to adjust for confounding when estimating both things.
>
> 2. Distribution of Expectations: We realize the lack of clarity regarding the distribution of expectations in the text. All our expectations are calculated concerning the observable distribution O=(X,A,E,T). We'll explicitly state this in the paper for better understanding.
>
> 3. Discrepancy in Bias vs. Outcome Regression (OR) and Relative RMSE: We display Relative RMSE because RMSE itself changes significantly over time, i.e., when we go from estimating the survival curve at time t=2 to time t=10, so we can’t really see the whole story on one set of axes. The choice to compare everything to OR gives us a reasonable scale to work with.
>
> 4. Assumptions Regarding Independent Censoring: Our method does not assume independent censoring, which sets it apart from some related works. We've highlighted in the covariate balancing section that our asymptotic bound is theoretically stronger and provably optimal even without this assumption compared to Xue et al. 2023. Additionally, we will try to clarify this further in the paper, emphasizing the strength of our approach in not relying on this assumption.
>
> We appreciate your constructive feedback, and these clarifications and adjustments will be incorporated to enhance the paper's clarity and completeness.

---

### Official Review · Reviewer_CLjR · 2023-10-30

**Soundness:** 3 good
**Presentation:** 3 good
**Contribution:** 3 good
**Rating:** 6
**Confidence:** 4

**Summary:**

The authors study the problem of estimating survival causal effects, where the aim is to characterize the impact of an intervention on survival times, i.e., how long it takes for an event to occur. This paper in general is interesting.

**Strengths:**

In causal survival effect estimation, classic estimators involve inverse probabilities are a product of treatment assignment and censoring probabilities. The authors propose a covariate balancing approach to estimating these inverses directly, sidestepping this problem.

**Weaknesses:**

I have a couple of comments:

The notation might not be fully rigorous. For example, P sometimes refers to probability, and sometimes refers to density.

Theorem 3.5 provides asymptotical linearity of $\hat \psi$ with balancing weights estimated by (19). Is the choice of (19) in some sense optimal? Could the authors comment on the variance of $\hat \psi$? Is it optimal/minimal?

A recently proposed causal survival forest (Estimating heterogeneous treatment effects with right-censored data via causal survival forests, Cui et al. 2023 JRSSB) can also be used to evaluate $\psi^{a,t}$. A comparison in simulations would be helpful.

By covariate balancing, Hirshberg & Wager (2021) are able to handle high dimensional data, can the proposed method deal with high-dimensionality?

**Questions:**

Please refer to the Weakness section.

---

> ### Author Response · Authors · 2023-11-23
> **Thank you for taking the time assess our paper and for pointing our potential confusing points. Based on your feedbacks, we have made the following clarifications and improvements.**
>
> 1. The probability P. We could not find in the manuscript where P has been wrongly attributed to density. All of our notations refer to probability because the space of time T, assignment A and event E are all discrete.
> 2. Optimality of the variance of $\hat \psi$. The variance of $\hat \psi$ is indeed optimal as we have mentioned after Theorem 3.5. To clarify this further, we will discuss more the connection to the influence function of $\hat \psi$. In essence, our method has similar asymptotic properties to other semi-parametric efficient methods for estimating the ATE because our Riesz representer estimator is a consistent estimator of the inverse propensity weight, as pointed out in section 3.
> 3. Causal Survival Forest. Thank you for pointing out this relevant paper. We will add the new experimental result.
> 4. Hirshberg & Wager (2021) able to handle high dimensional data. We chose to state our asymptotic results in nonparametric terms for simplicity, but high-dimensional asymptotic results analogous to those in Hirshberg & Wager (2021) follow from the same arguments.

---

### Official Review · Reviewer_ogxR · 2023-10-31

**Soundness:** 1 poor
**Presentation:** 1 poor
**Contribution:** 1 poor
**Rating:** 3
**Confidence:** 3

**Summary:**

This paper proposes a new way to estimate the causal effect in survival analysis.

**Strengths:**

The idea of this paper is interesting to investigate.

**Weaknesses:**

This paper is written in a very confusing way, which greatly reduces its value. Overall, the structure of the paper is confusing, the language is vague and uninformative, the proposed methodology is poorly described, and the technical details are not well explained. I list several specific points:
1. Abstract: the authors spend too much time discussing other things and make little mention of the proposed methodology and describes it vaguely.
2. Introduction: related work (section 4) should be moved to introduction.  I also recommend that authors only cite and discuss literature that is truly relevant/important to their work.
3. Setting: section 2.3 is really confusing. A lot of notation without explanation. It is more natural to introduce notation only when it needs to be used.
4. Approach: it reads like a stack of math techniques. Unimportant derivations or formulas can be omitted. More explanation would be helpful.
5. Experiment: the settings should be describe explicitly instead of citing other papers.

**Questions:**

1. Why do you focus on discrete survival analysis? Can your method be generalized to continuous time?
2. Is the asymptotic normality of your estimator examined in your experiments?

---

> ### Author Response · Authors · 2023-11-23
> **Thank you for dedicating time to assess our paper and provide invaluable feedback on the organization of our paper.**
>
> Thank you for dedicating time to assess our paper and provide invaluable feedback. We appreciate your insights into the presentation and structure of our work. We agree and have taken your suggestions seriously to enhance the readability of our paper.
>
> 1. Abstract. We acknowledge your recommendation and will revise our abstract as follows:
>
> We study the problem of estimating survival causal effects, where the aim is to characterize
> the impact of an intervention on survival times using debiased machine learning. Historically, parametric or semiparametric models (e.g. proportional hazards) were popular, but they come with problematic levels of bias. Recently debiased machine learning approaches become popular, especially in applications to large datasets. However, despite their appealing theoretical properties, these estimators tend to be unstable because the debiasing step involves the use of the inverses of small estimated probabilities. This problem is exacerbated in survival settings where propensities are a product of treatment assignment and censoring probabilities. We propose a covariate balancing approach to estimating these inverses directly, sidestepping this problem. The method works by estimating the propensities not by their functional forms but by their roles as the Riesz Representers of our linear functional parameter. The result is an estimator that is asymptotically optimal in theory and stable and efficient in practice. In particular, under overlap and asymptotic equicontinuity conditions, our estimator is asymptotically normal with negligible bias and optimal variance. Our experiments on synthetic and semi-synthetic data demonstrate that our method has competitive bias and smaller variance than debiased machine learning approaches.
>
> 2. Introduction and Related Work. We will make the following changes:
> Paragraph 3 in the introduction which talks about the advantages and the instability problem of previous debiased machine learning methods, will be streamlined, and will  incorporate paragraph 1 and 2 in related works.
> Paragraph 4 in the introduction which talks about covariate balancing will incorporate paragraph 3 in related works.
>
> 3. Section 2.3. We understand your concerns about the notation and its complexity in this section. These notations establish the framework for Section 3. We will move this subsection to section 3 and try to offer more contextual explanations for the readers.
>
> 4. Approach. We appreciate your feedback on this critical section. We firmly believe that the core innovation of our paper lies in the novel mathematical framework presented in Section 3. We will provide more contextual explanations and cut back on the theoretical details.
>
> 4. Experiments: We will bring the data generating process details from the Appendix to the main text.
>
> 5. Discrete time vs Continuous time SA: We chose to focus on discrete survival analysis due to its relevance in various real-world applications [1]. For example, In healthcare, patients may get checked up every few months so that we can not know exactly when the event happens, therefore discretized time is more robust. While our method primarily targets discrete survival analysis, we acknowledge the significance of continuous-time models. Theory-wise, the distinction between the 2 models, more specifically the fact that the discrete hazard is a conditional mean, while the continuous hazard is not, poses challenges in unifying the theoretical framework. We will tackle continuous SA in future work.
>
> 5. Asymptotic normality:
> Let Q be (estimate - estimand) / estimated standard error.
> The Kolmogorov-Smirnov statistics (the maximum difference between 2 CDFs) of Q produced by the methods vs. a normal distribution on the Synthetic dataset are: for Outcome Regression (OR): 0.93, for Double robust (DR): 0.25, for Balance: 0.06. We will also include a histogram of Q based on [2], which shows that Balance in fact approximates the normal distribution best.
>
> [1] Multivariate Survival Analysis and Competing Risk.
>
> [2] https://docs.doubleml.org/stable/guide/basics.html#regularization-bias-in-simple-ml-approaches

---

### Meta-Review · Area_Chair_7Ac9 · 2023-12-12

**Metareview:**

The authors study the problem of evaluating treatment effects expressed in terms of the survival function learned from right censored data. The population level treatment effect is expressed as an expectation over the hazards at time u. Instead of using the plug in estimate a first order approximation to the ATE is derived which requires estimation of the derivative (expressed as an inner product). To avoid instabilities that arise in inverse weighting, the paper proposes using the Riesz representer of the derivative resulting in an estimator that satisfies asymptotic efficiency. Most reviewers found the idea solid but expressed concerns about the organization of the work, clarity of the writing, empirical evaluation and comparisons to related work.

My personal opinion is that I think this paper contains a solid (and to my knowledge new) idea that deserves publication in a top ML conference; however I think the manuscript as it stands lacks polish does need revamping in terms of presentation, incorporation of baselines and improvements towards the accessibility of the ideas for a broader ML audience. In addition:

* One suggestion towards this is to highlight the key contributions of the work at the end of the introduction and bring up the related work (can keep as separate section) right after. This will help folks assess what this work brings to the literature relative to existing work early on in their reading.
* I am left uncertain by the author response to the question on (conditionally) independent censoring since (A1,A2), the assumptions for identification, explicitly assume a (conditionally) independent censorship process though the rebuttal states "we didn't assume independent censoring" -- in either case;
* The verification of asymptotic normality is nice to have but I do think the suggestion to compare to Cui et. al is sensible to incorporate in a revision since it does leverage orthogonal estimators to more robustly adjust for censorship.
* This is a minor point but the (legends, ticks, axes etc.) figures on page 9 are frankly illegible (when seen on a computer screen at arms length).

**Justification For Why Not Higher Score:**

This came down to a borderline decision for me (I've listed the reasons why in the review). If the changes are minor, I lean towards acceptance but here a new baseline needs to be run, there isnt clarity on whether or not the work assumes independent censorship and some work needs to be put in to improve the accessibility of the work. Overall, I think the idea is very good and above the bar for publication but the manuscript in its current form needs work.

**Justification For Why Not Lower Score:**

N/A

---

### Decision · Program_Chairs · 2024-01-16

Reject